# Antimicrobial Prescribing Confidence and Knowledge Regarding Drug Resistance: Perception of Medical Students in Malaysia and the Implications

**DOI:** 10.3390/antibiotics11050540

**Published:** 2022-04-19

**Authors:** Mainul Haque, Tasim Ara, Md. Ahsanul Haq, Halyna Lugova, Siddhartha Dutta, Nandeeta Samad, Abdullahi Rabiu Abubakar, Sharifah Shasha Binti Syed Mohdhar, Md. Mahabubur Rahman, Salequl Islam, Nihad Adnan, Rahnuma Ahmad, Shahidah Leong Binti Abdullah, Mohd Hafizi Bin Ismail, Brian Godman

**Affiliations:** 1Unit of Pharmacology, Faculty of Medicine and Defence Health, Universiti Pertahanan Nasional Malaysia (National Defence University of Malaysia), Kem Perdana Sungai Besi, Kuala Lumpur 57000, Malaysia; 2Institute of Statistical Research and Training, University of Dhaka, Dhaka 1000, Bangladesh; tara@isrt.ac.bd (T.A.); mrahman4@isrt.ac.bd (M.M.R.); 3Gonoshasthaya-RNA Molecular Diagnostic & Research Center, Dhanmondi, Dhaka 1205, Bangladesh; ahsan@rnabiotech.com.bd; 4Unit of Community Medicine, Faculty of Medicine and Defence Health, National Defence University of Malaysia Kuala Lumpur 51000, Malaysia; halina@upnm.edu.my; 5Department of Pharmacology, All India Institute of Medical Sciences, Rajkot 360001, Gujarat, India; siddhartha.dutta87@gmail.com; 6Department of Public Health, North South University, Dhaka 1229, Bangladesh; nandeeta.samad@northsouth.edu; 7Department of Pharmacology and Therapeutics, Faculty of Pharmaceutical Sciences, Bayero University, PMB 3452, Kano 700233, Nigeria; unisza7@gmail.com; 8Unit of Internal Medicine, Faculty of Medicine and Defence Health, Universiti Pertahanan Nasional Malaysia (National Defence University of Malaysia), Kem Perdana Sungai Besi, Kuala Lumpur 57000, Malaysia; shashamohdhar@gmail.com; 9Department of Microbiology, Jahangirnagar University, Savar, Dhaka 1342, Bangladesh; salequl@gmail.com (S.I.); nihad@juniv.edu (N.A.); 10Department of Physiology, Medical College for Women and Hospital, Plot No 4 Road-8/9, Sector-1, Dhaka 1230, Bangladesh; rahnuma.ahmad@gmail.com; 11Unit of Military Medicine, Faculty of Medicine and Defence Health, Universiti Pertahanan Nasional Malaysia (National Defence University of Malaysia), Kem Perdana Sungai Besi, Kuala Lumpur 57000, Malaysia; shahidah2013@gmail.com; 12Unit of Administration, Faculty of Medicine and Defence Health, Universiti Pertahanan Nasional Malaysia (National Defence University of Malaysia), Kem Perdana Sungai Besi, Kuala Lumpur 57000, Malaysia; hafizi@upnm.edu.my; 13Department of Pharmacoepidemiology, Strathclyde Institute of Pharmacy and Biomedical Sciences, University of Strathclyde, Glasgow G4 0RE, UK; brian.godman@strath.ac.uk; 14Centre of Medical and Bio-allied Health Sciences Research, Ajman University, Ajman 13306, United Arab Emirates; 15Division of Public Health Pharmacy and Management, School of Pharmacy, Sefako Makgatho Health Sciences University, Pretoria 0208, South Africa

**Keywords:** antimicrobials, prescribing, antimicrobial drug resistance, knowledge, perception, medical students, Malaysia

## Abstract

Background: Worldwide, microbes are becoming more challenging by acquiring virulent skills to adapt and develop antimicrobial resistance (AMR). This is a concern as AMR increases morbidity, mortality, and costs. Consequently, physicians need to be trained on appropriate antimicrobial prescribing, starting as medical students. Objective: To evaluate medical students’ confidence in antimicrobial prescribing and AMR. Methods: Cross-sectional study assessing medical students’ knowledge, perception, and confidence in prescribing antimicrobials and AMR in a Malaysian University. A universal sampling method was used. Results: Most responding students believed that educational input regarding overall prescribing was sufficient. Regarding the principle of appropriate and accurate prescriptions, female medical students had less knowledge (odds ratio (OR) = 0.51; 95% confidence interval (CI) 0.25–0.99; *p* = 0.050). Year-IV and Year-V medical students had more excellent knowledge than Year-III students regarding confidence in potential antibiotic prescribing once qualified. Year-V students also showed an appreciably higher confidence in the broad principles of prescribing, including antibiotics for infectious diseases, compared to those in other years. Conclusion: Overall, medical students gain more knowledge and confidence regarding the potential prescribing of antimicrobials as their academic careers progress. This is important given concerns with the current excessive use of antimicrobials in Malaysia.

## 1. Introduction

Microorganisms, including bacteria, viruses, fungi, and other parasites, have increasingly adapted and become resistant to commonly prescribed antimicrobials [1,2]. Drug resistome is a vigorous and escalating public-health concern [3]. The resistome encompasses all antimicrobial resistance (AMR) genes and contains resistance genetic components found equally among pathogens and antimicrobial-producing microbes, with enigmatic resistance genes present in microbial chromosomes [4]. Increasing AMR is fueled by excessive and inappropriate use of antimicrobials [5,6,7,8]. Other factors increasing AMR include overpopulation, excessive self-purchasing of antibiotics without a prescription for self-limiting conditions including upper respiratory tract infections, traveling, water pollution, and a lack of hygiene and water sanitation [3,9,10,11,12].

This is a concern as AMR increases morbidity, mortality, and costs [13,14,15]. There were an estimated 4.95 million deaths worldwide in 2019 associated with bacterial AMR, highest in sub-Saharan Africa and South Asia, and rising [16]. The World Bank believes the annual worldwide GDP would decrease by 1.1% in a low-impact AMR scenario by 2050, potentially up to 3.8% in a high-impact AMR scenario and possibly over 5% [13]. As a result, the cost of dealing with the consequences of AMR could possibly increase from $300 billion per year currently to more than $1 trillion per year by 2050 [14,17].

Consequently, there is an urgent need for coordinated approaches to combat rising AMR rates and the implications, especially among low- and middle-income countries (LMICs), which include improved antimicrobial stewardship programs (ASPs) [16,18,19,20]. Microbes can easily cross geographical borders, with resistant microbes affecting both high-income countries (HICs) as well as LMICs, although more significant in LMICs [16,20]. The growing awareness of the clinical and economic costs of AMR have resulted in global and national action plans to reduce AMR coordinated by the WHO and others [16,20,21,22]. This includes Malaysia, which has growing AMR rates and concerns with appropriate antimicrobial prescribing across sectors [20,23,24,25,26,27,28]

Improving medical students’ knowledge regarding antibiotics and AMR are essential to enhance appropriate antimicrobial prescribing following graduation, achieved through ASPs and other programs [29,30,31,32]. A comprehensive understanding of the pathophysiology of diseases as well as clinical pharmacology and therapeutics, are also indispensable to improve their prescribing practices post-qualification along with quality targets [33,34,35,36]. Moreover, any healthcare professionals’ (HCP) purpose of treating a patient should be documented in their notes and duly signed, ideally in line with current prescribing guidance [36,37]. Their confidence with prescribing decisions will be enhanced by appropriate training at university.

In both public and private primary health care settings in Malaysia and emergency departments, antimicrobials are often prescribed for self-limiting diseases including upper respiratory tract infections (URTIs) [27,28,38,39,40]. However, this was more noticable in the private health care system, potentially enhanced by financial considerations including fees from dispensing as well as more significant patient pressures [28,38,41]. Long waiting times to see HCPs in primary healthcare clinics (PHCs) in Malaysia further adds to the pressure on HCPs to prescribe antimicrobials rather than spend valuable time providing an explanation why they are reluctant to prescribe [14,27,32,42,43,44,45]. This is a concern given rising AMR rates in Malaysia [24,46,47,48,49,50,51]. HCPs, including physicians, are a key stakeholder group to target as they can appreciably influence antimicrobial utilization patterns in Malaysia. This is similar to other LMICs, with prescribing being a significant activity among ambulatory care physicians [52]. Encouragingly, multiple interventions, including educational interventions, can reduce unnecessary prescribing of antibiotics for essentially viral infections, including URTIs, among physicians in LMICs [14,32,53]. This should be borne in mind during undergraduate teaching and followed-up postqualification.

However, it has been reported that Malaysian university students, including medical students, often take antimicrobials without any prescription [54]. This is despite such activities being against current regulations; however, this still happens among the general population [54,55,56]. In addition, there are ongoing concerns across countries, including Malaysia, that inappropriate dispensing of antibiotics without a prescription appreciably adds to growing AMR rates [14,32,54,57]. Earlier studies also showed that whilst medical students in Malaysia possess a sound knowledge about prescribing, including antimicrobials, they felt there was a gap between theoretical and practical clinical pharmacology input. In view of this, they suggested additional teaching-learning hours regarding prescribing skills, including antimicrobials, should be built into the curriculum [58,59]. This is because inadequate knowledge can enhance irrational prescribing post-qualification, which is challenging to redress. Consequently, it is better to intervene before inappropriate prescribing habits start to develop during studentship before graduation [58,59], rather than wait post-qualification when poor prescribing habits have become ingrained.

Consequently, we wanted to build on these earlier findings in Malaysia to provide future guidance on the curriculum for medical students going forward. Because of this, this study sought to determine Malaysian medical students’ self-confidence and knowledge in managing infectious diseases and appropriate antimicrobial prescribing, and correlating this with proficiency regarding infectious diseases management, during their clinical training. This study also sought to assess the relative effectiveness of various instruction/teaching delivery modes in gaining and retaining knowledge of medicines and prescribing including antimicrobial prescribing. The combined findings can help improve physician education in Malaysia and broaden it in the future with increasing challenges from viral and other diseases.

## 2. Materials and Methods

### 2.1. Study Design

A cross-sectional study was undertaken to assess medical students’ knowledge and perception of prescribing antimicrobial agents and AMR during their clinical training. A survey questionnaire was employed as a data collection tool among Year-III–V medical students.

### 2.2. Study Population

The study population was Year-III–V medical students of the Faculty of Medicine and Defence Health, Universiti Pertahanan Nasional Malaysia ((UPNM) the National Defence University of Malaysia), Kem Perdana Sungai Besi, Kuala Lumpur, Malaysia. This study comprised three ethnic groups: Malay, Chinese, and Indian. There are currently three categories of medical students admitted to UPNM. Those are cadet officers, territorial army, and civilians.

### 2.3. Study Period

The data collection exercise was undertaken between 7 January 2018 and 21 March 2019, i.e., before the COVID-19 pandemic with its restrictions on university education [60,61,62,63]. Year-III clinical students were initially unavailable on campus during the initial data collection period. Similarly, other students were away at various times for clinical activities necessitating an extended period for data collection.

### 2.4. Sampling Method and Sample Size

The survey was conducted using a universal sampling method comprised of all Year-III, Year-IV, and Year-V medical students from 2018 to 2019 academic sessions. The research group distributed 155 study instruments among clinical Year-III, Year-IV, and Year-V students during the principal study, and 15 were dispersed during the pretested phase. A total of 170 study instruments were distributed.

### 2.5. Data Collection Tool (Questionnaire)

The data were collected using a validated questionnaire on antimicrobial prescribing knowledge, and perception was determined based on a previous study conducted by Weier et al. (2017) [64]. The study instrument was subsequently pretested and validated for the local context in Malaysia. Five medical students in each of the clinical years (5 × 3 = 15) participated in the questionnaire validation process. They did not participate in the principal study.

The questionnaire comprised six (6) sections, A to F: section A: demographic information; section B: sufficiency of education and confidence in their knowledge; section C: modes of teaching and confidence in clinical situations; section D: perceptions of antimicrobial resistance (AMR); section E: knowledge of prescribing guidelines; section F: demonstration of clinical knowledge. Demographic details included gender, ethnicity, and year of study. This is because studies undertaken among students in Malaysia as well as physicians from other countries have shown these factors impact on antimicrobial prescribing and AMR [16,32,58,65,66,67].

### 2.6. Survey Reliability

A reliability analysis of the survey tool was undertaken. The Cronbach’s alpha obtained was 0.9 for questions relating to the sufficiency of education, confidence in knowledge in different subject areas, confidence in various clinical situations, and perceptions of AMR.

### 2.7. Ethical Approval

This research was reviewed and approved by the Institutional Research Ethical Committee from the Centre for Research, Innovation and management, National Defense University, Malaysia (code: UPNM/2019/SF/SKK/04; reference number: UPNP (PPPI) 16.01/06/024 (2), dated 3 January 2019). The participation in this study was completely voluntary and anonymous. Before distributing the instrument, the researchers clearly explained the aim, scope, and future potential issues before every data collection occasion. A printed information sheet was also provided to each student to learn more about the study. Additionally, researchers obtained written consent (approval) before research respondents participated in the pretest and the principal study.

### 2.8. Data Analysis

The data were entered into an Excel file, which was then transferred into SPSS-22 software (IBM SPSS Statistics for Windows, version 23.0. IBM Corp., Armonk, NY, USA) and with Stata 15 (StataCorp, LP, College Station, TX, USA) for further analysis and the graphs were prepared with GraphPad prism 8.3.0. The significance level was established at *p* ≤ 0.05. Pearson’s chi-square tests were used to determine the relationship between confidence level in infectious disease knowledge and other variables. A logistic regression model was introduced to explore the predictors of a confident group. Initially, the univariate logistic regression technique analyzed the relationship between several predictors and confidence levels in infectious diseases. Exploratory variables including gender, years of the students, e.g., Years III, IV and V, their ethnicity, and designation in infectious diseases were regressed onto their confidence in infectious diseases knowledge as a response variable. An odds ratio (OR) greater or less than 1 indicated a greater probability or lower probability of being confident in the knowledge of infectious diseases compared to the reference category. Confidence in antibiotic prescribing and knowledge and attitude towards AMR score differences among explanatory variables were assessed by an independent sample *t*-test.

## 3. Results

The demographic distribution of the clinical students is contained in Table 1. Among the enrolled students, 47.1% and 52.9% were male and female, respectively. Out of the 140 research respondents, 47.9% were 22 years of age, and 68.6% were Malay. There were 47.9%, 39.3%, and 12.9% civilians, cadet officers, and territorial army students, respectively. The research respondents were from Years III, IV, and V. The response rate was 90.32%. The total study population was 170 (15 (pretest) + 140 (principal study) + 5 (discarded because incomplete data) + 10 (study instrument were not returned)).

### 3.1. Sufficiency in the Level of Education and Confidence in Knowledge Regarding Prescribing among Students

The majority of the clinical medical students (males = 49.2% and females = 50.9%) felt that their teaching-learning was sufficient for commonly prescribed drugs (Table 2). Similarly, UPNM medical students (males = 52.7% and females = 47.8%) agreed that sufficient formal education and training is currently being provided regarding appropriate and accurate prescription writing principles. Correspondingly, 48.8% of males and 51.2% of females believed their formal education, training, and pharmacological confidence was sufficient (Table 2). Overall, half of the respondents generally agreed that among the key sections, i.e., (i) commonly prescribed drugs, (ii) principles of appropriate and accurate prescription writing, and (iii) infectious diseases, sufficient educational input was given. However, there were statistically significant differences (Table 2) between the genders in appropriate and accurate prescription writing (*p* = 0.050). Nevertheless, most medical students had confidence in their knowledge level regarding these issues.

Regarding the principles of appropriate and accurate prescription writing, female medical students showed less knowledge (odds ratio (OR) = 0.51; 95% confidence interval (CI) 0.25–0.99; *p* = 0.050) compared to males (Table 2 and Figure 1). Year-V students had an 8.0 times higher knowledge level (95% CI 1.05–67.4; *p* = 0.049) than Year-III students (Table 3). All three ethnic groups were overall satisfied regarding their educational input, commonly prescribed drugs (80.2%), principles of appropriate and accurate prescription writing (64.6%), and infectious diseases (89.6%). No statistically significant difference was observed among the ethnic groups in the three components of the formal education and training on pharmacology (Table 4).

### 3.2. Modes of Teaching and Confidence in Clinical Situations

Both male and female medical students believed that all five modes of teaching, i.e., lectures (89.4%), tutorials/workshops/problem-based learning (PBL) sessions (90.9%), clinical rotations (98.5%), informal education by fellow residents and registrars (87.9%), and attending patient care rounds/clerking/ward rounds (97.0%) were helpful or effective in attaining and improving knowledge about medicine and prescribing (Table 5). There were no statistically significant differences in the mode of teaching and knowledge and prescribing, including antimicrobials. Additionally, there were no statistically significant differences among the genders and the study years concerning the ways of instruction. Year-III, Year-IV, and Year-V medical students were generally satisfied and valued all five different instructional methods (Table 6).

Furthermore, among all three ethnic origins, the majority (84%) stated their trust, usefulness, and satisfaction regarding the five different teaching methods.

### 3.3. Confidence in Antimicrobial Prescribing

In every aspect of confidence with potential antibiotic prescribing, Year-IV and Year-V students had higher knowledge than Year-III students. Additionally, Year-V students had significantly more knowledge than Year-IV students (Figure 2).

**Figure 2 antibiotics-11-00540-f002:**
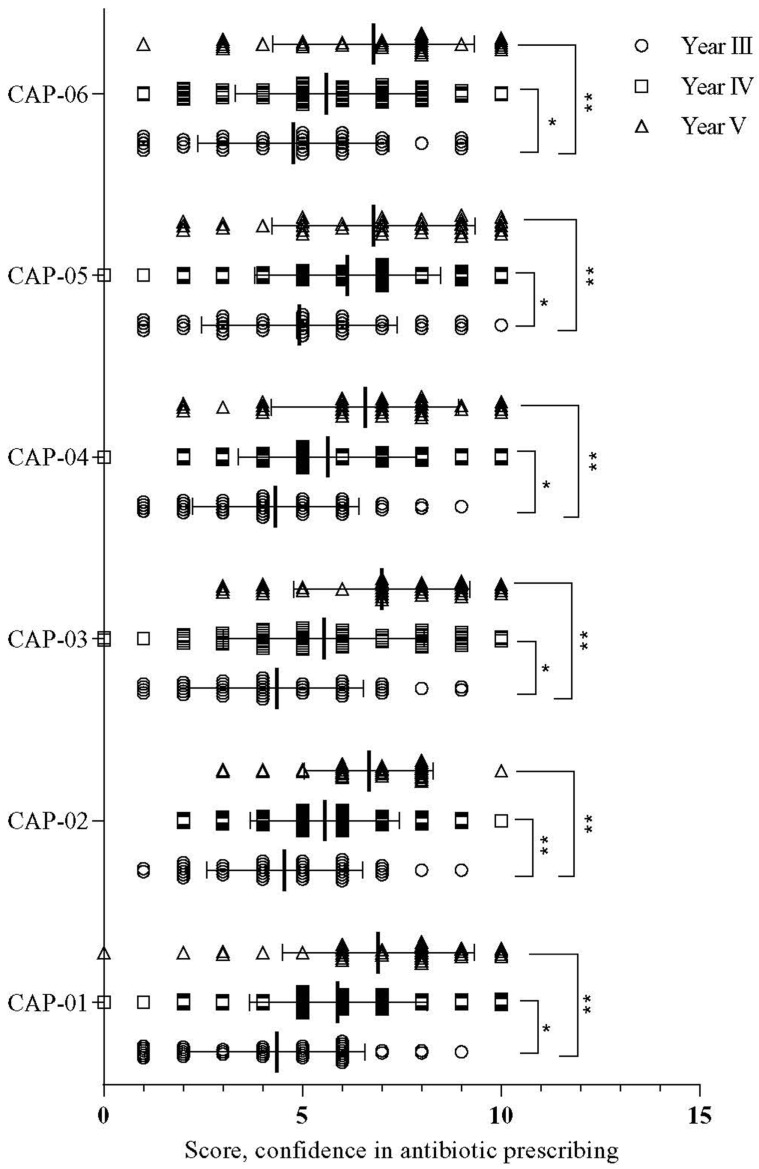
Mean difference of confidence in antibiotic prescribing (CAP) among students of different years. Note: ** *p* = 0.002–0.09 and * *p* < 0.05. CAP-01: accurately diagnosing community-acquired pneumonia; CAP-02: accurately interpreting pathology and microbiology results; CAP-03: knowing the right regimen (dose, frequency, and route of administration) with antibiotic treatment for a specific indication such as pneumonia or an exacerbation of COPD; CAP-04: knowing the right duration for antibiotic treatment for a specific indication such as pneumonia or an exacerbation of COPD; CAP-05: identifying situations where antibiotic treatment is not necessary; CAP-06: knowing when antibiotic treatment needs to be adjusted, stopped, or other treatments need to be prescribed. Circle, square, and triangles indicate Year-III, Year-IV, and year-V, respectively. While the relative effectiveness of various modes of delivery in gaining and retaining the knowledge of medicines and prescribing was compared with race no significant association was noted (Table 7). Majority student respondent that it was useful to them.

**Table 7 antibiotics-11-00540-t007:** Frequency and odds of student years in the relative effectiveness of various delivery modes in gaining and retaining the knowledge of medicines and prescribing, including but not limited to antibiotics prescribing.

	Malay	Chinese	Indian
Lectures			
Least useful	9 (9.38%)	1 (7.69%)	5 (16.1%)
Useful	87 (90.6%)	12 (92.3%)	26 (83.9%)
Tutorials or workshops of PBL sessions			
Least useful	9 (9.38%)	1 (7.69%)	5 (16.1%)
Useful	87 (90.6%)	12 (92.3%)	26 (83.9%)
Clinical rotations			
Least useful	1 (1.04%)	2 (15.4%)	0
Useful	95 (99.0%)	11 (84.6%)	31 (100%)
Informal teaching by fellow residents and registrars			
Least useful	7 (7.29%)	2 (15.4%)	3 (9.68%)
Useful	89 (92.7%)	11 (84.6%)	28 (90.3%)
Attending patient care rounds/ward rounds			
Least useful	3 (3.135)	2 (15.4%)	1 (3.23%)
Useful	93 (96.9%)	11 (84.6%)	30 (96.8%)

NB: Data were presented as the percentage number in the parenthesis.

### 3.4. Comparative Confidence in Antimicrobial Prescribing

Year-III and Year-IV students showed the same confidence level in all three components (pharmacology knowledge, principles in prescribing, and infectious diseases). Year-V students showed 4, 5.6, and 5.9 times higher confidence than Year-III students in knowledge regarding the principles in prescribing (95% CI 1.51–11.02; *p* = 0.006) and prescribing knowledge for infectious diseases (95% CI 1.95–22.2; *p* = 0.002) (Table 8 and Figure 3) respectively. Malay and Chinese students showed the same levels of confidence; however, Indian students had more knowledge and confidence regarding pharmacology (OR = 1.72; 95% CI 1.17–6.23; *p* = 0.019) and infectious diseases (OR = 3.42; 95% CI 1.21–9.68; *p* = 0.021) (Table 9 and Figure 4). The prescribing confidence in different clinical situations is depicted in Figure 5.

### 3.5. Knowledge and Attitudes towards AMR

Almost all knowledge and attitude towards AMR scores were lower among Year-III students than Year-IV and Year-V students (Figure 6). The perceptions of medical students regarding the different factors regarding AMR are illustrated in Figure 7.

### 3.6. Knowledge of Prescribing Guidelines

The students felt that the trust in antimicrobial prescribing guidelines and adherence to these guidelines was essential to reduce the risk of AMR. The cadet officer medical students showed 2.46 times the odds (95% CI 1.12–5.42, *p* = 0.025) of the civilians (Figure 8A) regarding the awareness of the guidelines available in Malaysia. The Indian students had a lower knowledge than the Malay students (OR = 0.21, 95% CI 0.06–0.76; *p* = 0.017). No other significant differences were noted for other guidelines (Figure 8B).

## 4. Discussion

The response rate (90.32%) was relatively high. This may be because research respondents were from a military school with strict discipline alongside a small study population with good student cooperation.

Encouragingly, most students believed that the Faculty of Medicine and Defence Health provided sufficient teaching-learning educational inputs regarding commonly prescribed medicines including antimicrobials, appropriate and accurate prescription writing principles, and the management of infectious diseases. This is welcome as an earlier study conducted among medical students in Malaysia requested more educational input regarding antimicrobial prescribing and clinical pharmacology [59]. Our findings are similar to several other studies researching educational input among medical students regarding antibiotics and AMR [64,68,69,70,71,72]. A previous study in Malaysia also showed that most medical students possessed a reasonably good knowledge of antimicrobials [54]. This study also showed that both male and female medical students had confidence in appropriate and accurate prescription writing [54].

However, others have reported different findings [52,59,73]. In India, Nayak et al. (2021) found that most of the medical students surveyed felt that the current input regarding pharmacology was not sufficient to develop appropriate prescribing skills [52]. Similarly, a study conducted in South Africa reported a low level of prescribing confidence among students, which resulted in greater input on clinical pharmacology and prescribing during undergraduate teaching [73]. One study conducted among both University nonmedical and medical students in Malaysia reported that their overall knowledge was low regarding antimicrobials and other health-related issues; however, medical students’ knowledge levels were significantly (*p* < 0.001) higher regarding antimicrobials and other matters than nonmedical participants [74]. Higuita-Gutiérrez et al. (2020) also found that a significant proportion of medical students in their study in Colombia found the training on antibiotics and AMR to be mediocre to poor [75]. Similarly, in their systematic review, Nogueira-Uzal et al. (2020) also found a considerable lack of knowledge regarding antibiotics in the included studies, with 41–69% of medical students in the various studies believing antibiotics would help to treat URTIs [76]. This is a concern especially given high inappropriate prescribing of antibiotics in ambulatory care particularly across LMICs and including Malaysia [8,14,27,28,32].

The finding that Year-V medical students had eight times higher knowledge levels regarding commonly prescribed drugs versus Year-III students is a concern going forward that needs to be addressed. However, this is different from studies conducted in South Africa and in Spain, where there were low confidence levels regarding the prescribing antibiotics among final year medical students [68,73]. The Spanish study found that whilst medical students felt confident in diagnosing infectious diseases, they needed more education and training regarding judiciously prescribing antimicrobials [68]. Haque et al. (2016) also found that medical students in Malaysia welcomed more education on the selection of antibiotics, perceiving a gap between theoretical input and clinical practice [59]. This was similar to the situation in India and across Europe, especially among several Central, Eastern, and Southern European countries, where there were concerns with knowledge about antibiotics, prescribing skills for common infectious diseases, and AMR, with a need for additional educational input [77,78,79,80]. A further study conducted among medical students in France and Sweden found that a multimodal instructional strategy resulted in better prescribing habits among students versus being exposed to a lower number of teaching-learning methods, which has important implications [78].

Our findings also demonstrate that students’ capability to prescribe independently and rationally during clinical rotations improves with training, which is encouraging. This contrasts with other studies showing that medical students require more teaching-learning sessions on antimicrobials prescribing for their forthcoming practice post-qualification [32,59,77,78,79,80,81,82,83,84,85]. We are not sure of the reasons behind this difference. This may reflect differences in teaching approaches, including clinical rotation programs between countries with problem-based learning styles associated with greater knowledge about antibiotics and prescribing than traditional learning approaches [80]. Of interest when refining the content of teaching modules for medical students, we found that Year-III and Year-IV students had the same level of confidence in all four components. However, Year-V students exhibited around four to six times higher confidence levels than Year-III students. This is important given the appreciable changes in the education of HCPs across countries as a result of the COVID-19 pandemic and the implications on university education especially in LMICs [86,87,88].

Furthermore, AMR knowledge and attitude scores were typically lower among Year-III medical students than among their seniors. It is evident that as students reach a senior level, their understanding and skills improve, especially with additional input at a senior level [89,90]. In any event, teaching prudent antibiotic prescribing skills and AMR is essential to reduce rising AMR rates [18,91].

Encouragingly, both genders in the study found all five modes of teaching were helpful or effective in attaining and improving knowledge about medicine and prescribing. The same situation was seen regarding knowledge of antimicrobials and their prescribing. Alongside this, there were no statistically significant differences regarding the modes of teaching among the genders, years of study, i.e., Years III, IV and V, and ethnic origin. The exception was with Chinese and Indian medical students in certain aspects. Chinese students felt less benefit in the clinical rotation educational method than other ethnic groups. On the other hand, Indian medical students appeared to have higher levels of understanding than those from other ethnic origins. We are not sure of the reasons behind these differences and will be exploring them further in future studies

Encouragingly, the medical students in this study showed awareness about the prescribing guidelines available in Malaysia to assist with appropriate disease management and enhance prudent/rational prescribing. This is important as multiple publications have concluded that adherence to guidelines, hospital formularies, and other strategies does improve prescribing quality, including antimicrobials [14,32,35,36,92,93,94].

## 5. Limitation of the Study

This study was cross-sectional with its inherent limitations. There were also issues with data collection as a result of clinical placements. This impacted the final classification of some students, e.g., some of the Year-V students were wrongly labelled as Year-IV. The research was only conducted at only one university in Malaysia and before the forced closure of universities due to the COVID-19 pandemic. Despite these limitations, we believe the findings will be of interest to other countries and provide guidance to the universities in Malaysia.

## 6. Conclusions

The majority of the clinical medical students of UPNM reported they received sufficient input regarding commonly prescribed medicines, including antimicrobials, appropriate and accurate prescription writing principles, and infectious diseases. Year-V students had an eight times higher knowledge level regarding commonly prescribing medicines than Year-III students and appreciably higher confidence in the principles of prescribing antimicrobials in patients with infectious diseases. Year-IV and Year-V students were also found to have a better understanding of AMR. Generally, as students became more senior, their knowledge, attitude, and understanding regarding infectious disease and AMR improved together with their future prescribing skills and confidence including antimicrobials prescribing.

## 7. Recommendation

There is a need for medical students to be familiarized with applying rational approaches to prescribing, including antimicrobials prescribing, and these should be included earlier in the curriculum. This is even more important due to the pandemic and the resultant increased prescribing of antimicrobials for essentially viral infections. In several LMICs countries, clinical clerking and teaching, including prescribing skills, has especially been impaired because of lockdown and movement control orders. These challenges need to be addressed at the earliest possible opportunity to ensure AMR does not become the next pandemic to address.

## 8. Article Highlights

This study determined the self-confidence and knowledge of Malaysian medical students of different clinical years in managing infectious diseases and prudent antimicrobial prescribing.

Students reported didactic lectures as the least helpful teaching method and clinical rotation as the most beneficial.

The medical students of different clinical years surveyed possessed knowledge sufficiency and confidence regarding infectious diseases.

The majority of the medical students knew the guidelines to assist with appropriate antibiotic prescribing.

This study revealed that medical students reach senior years knowing and having prescribing confidence of antimicrobial medication advances.

## Figures and Tables

**Figure 1 antibiotics-11-00540-f001:**
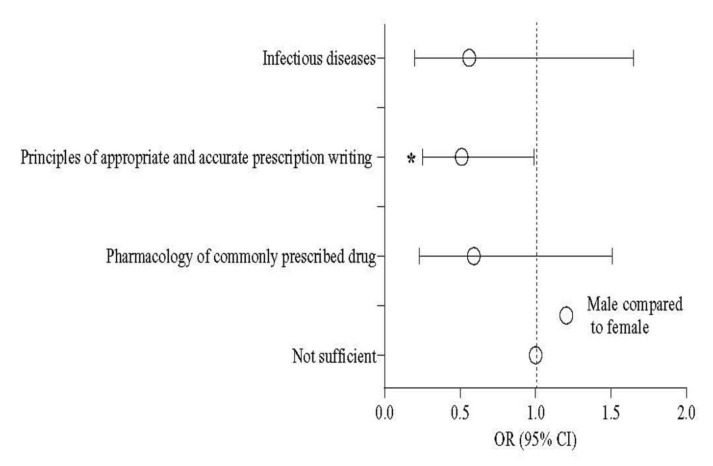
Odds of gender in the formal education and training on pharmacology. **Note**: * *p* < 0.05. Circles indicate the estimation of odds ratio.

**Figure 3 antibiotics-11-00540-f003:**
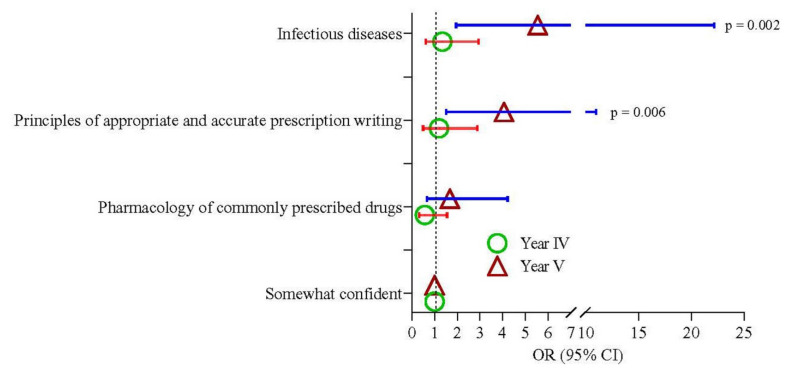
Odds of confidence in knowledge among the Year-IV and Year-V medical students compared to Year-III. Triangle and circle denote the estimation of odds ratios of Year-IV and Year-V student, respectively.

**Figure 4 antibiotics-11-00540-f004:**
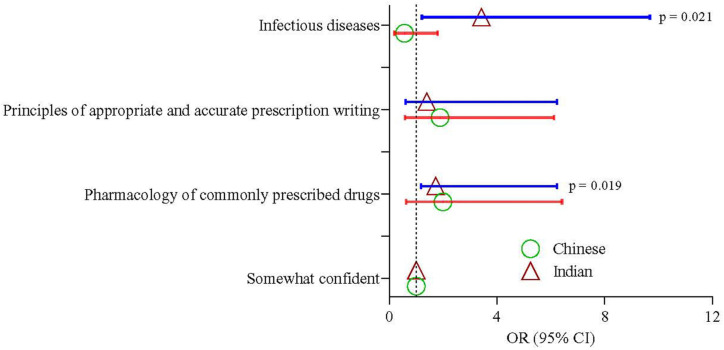
Odds of confidence in knowledge among the Chinese and Indian students compared to the Malay students. Triangle and circle represent the estimates of odds ratios of Indian and Chinese student, respectively.

**Figure 5 antibiotics-11-00540-f005:**
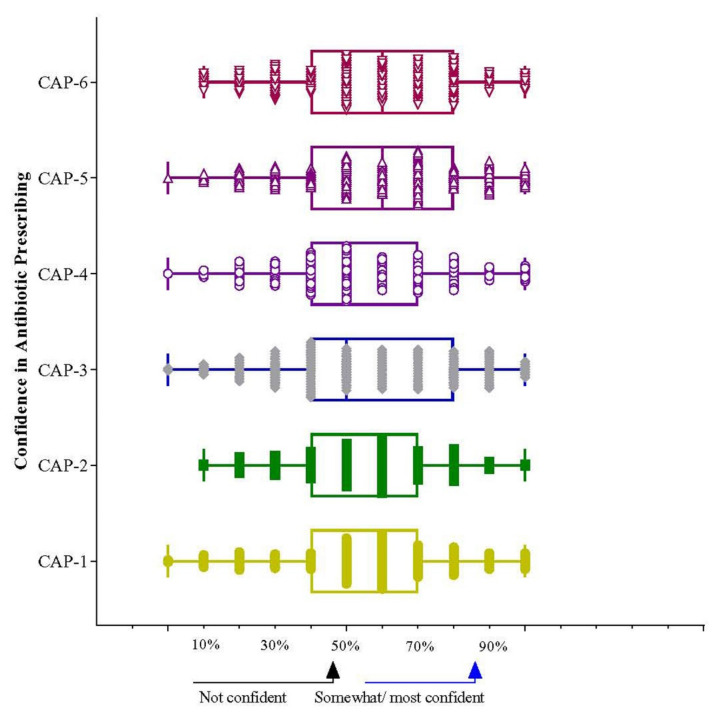
Scores of confidence in knowledge in different clinical situations. Data are presented as percentiles where <50% means not confident whereas >50–70% indicates somewhat and >70% indicates most confident. Note: CAP-01: accurately diagnosing community-acquired pneumonia; CAP-02: accurately interpreting pathology and microbiology results; CAP-03: knowing the proper regimen (dose, frequency, and route of administration) for antibiotic treatment for a specific indication such as pneumonia or an exacerbation of COPD; CAP-04: knowing the suitable duration for antibiotic therapy for a particular indication such as pneumonia or a worsening of COPD; CAP-05: identifying situations where antibiotic treatment is not necessary; CAP-06: knowing when antibiotic treatment needs to be adjusted, stopped, or other treatments prescribed.

**Figure 6 antibiotics-11-00540-f006:**
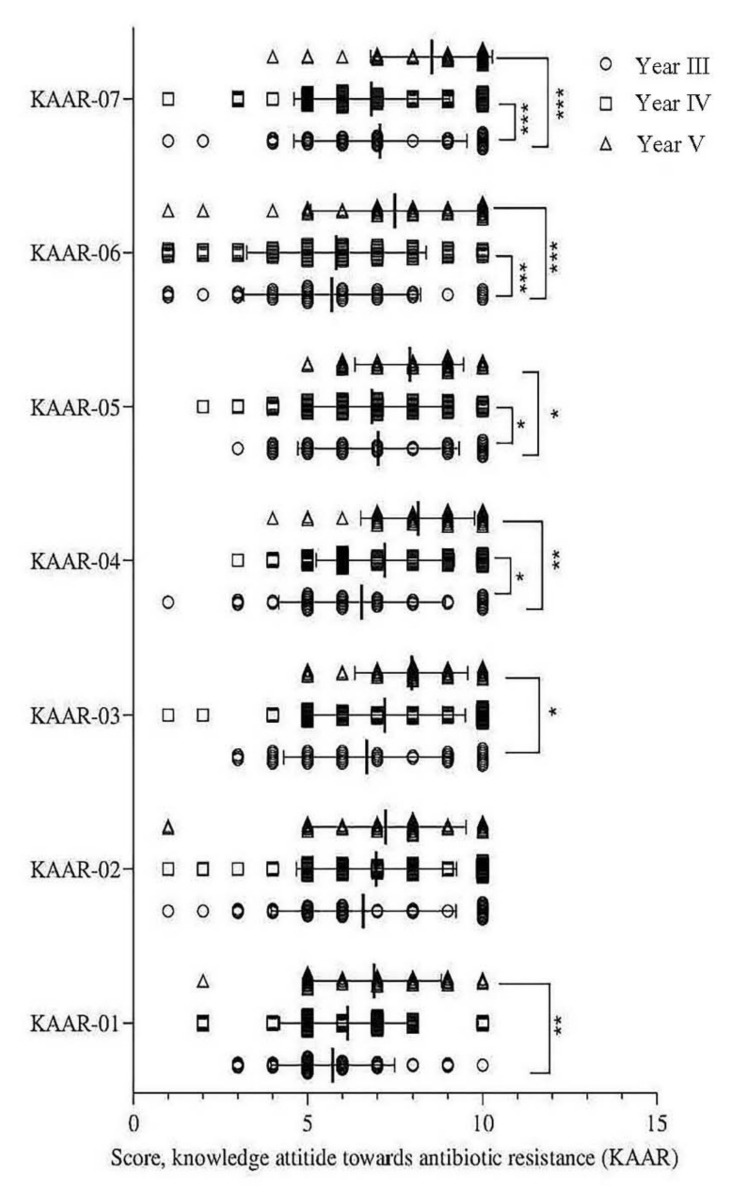
Mean difference of knowledge attitude towards antibiotic resistance (KAAR) among students of different years. Data are presented as mean with standard deviation after plotting all the observations in the chart. The comparison was made between Year-III, Year-IV and Year-V students. **Note**: *** *p* < 0.001; ** *p* = 0.002–0.09 and * *p* < 0.05. KAAR-01, few antibiotics being developed; KAAR-02, prescribing antibiotics when the situation does not warrant their use; KAAR-03, using the wrong antibiotic for the situation; KAAR-04, using an inappropriate dose and/or frequency of antibiotic for the situation; KAAR-05, using antibiotic treatments for a longer duration than indicated; KAAR-06, not prescribing antibiotics when the situation requires their use; KAAR-07, patient noncompliance with antibiotic treatment (such as not taking it as prescribed, not completing the course, or taking too much). Circle, square, and triangles indicate Year-III, Year-IV, and Year-V, respectively.

**Figure 7 antibiotics-11-00540-f007:**
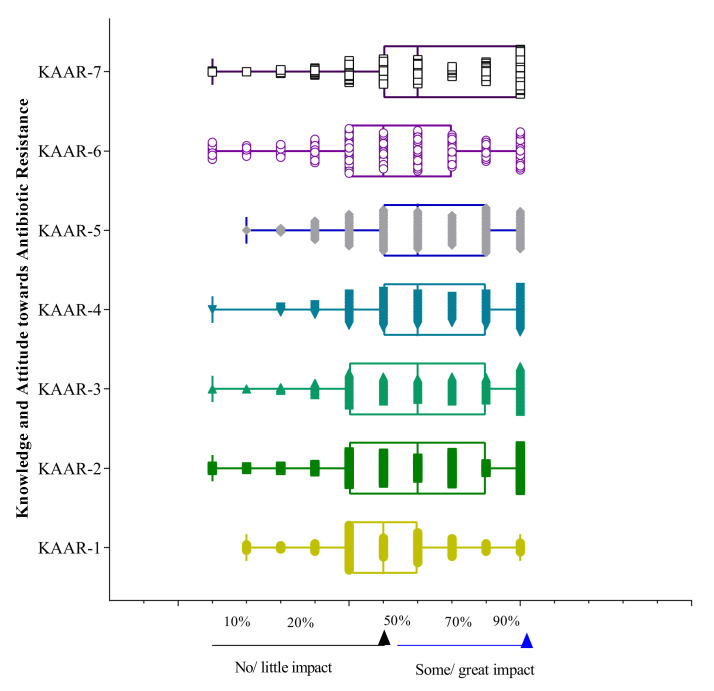
Scores of perceptions of knowledge and attitude towards antibiotic resistance, and impact of different factors have on antimicrobial resistance. Data are presented as percentiles where <50% means no or little impact whereas >50–70% indicates some impact and >70% indicates a great impact. Note KAAR-01, few antibiotics being developed; KAAR-02, prescribing antibiotics when the situation does not warrant their use; KAAR-03, using the wrong antibiotic for the situation; KAAR-04, using an inappropriate dose and/or frequency of an antibiotic for the situation; KAAR-05, using antibiotic treatment for a longer duration than indicated; KAAR-06, not prescribing antibiotics when the situation requires their use; KAAR-07, patient noncompliance with antibiotic therapy (such as not taking the medication as prescribed, not completing the course, or taking too much).

**Figure 8 antibiotics-11-00540-f008:**
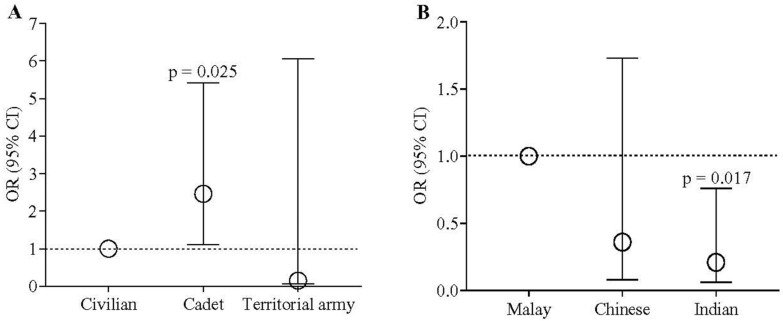
Odds of designation and ethnicity of the participants regarding the knowledge of prescribing guidelines. Logistic regression was used to estimate the *p*-value. Circles indicate the estimate of odds ratio.

**Table 1 antibiotics-11-00540-t001:** Demographic distribution of the data.

Variables	Response (n = 140)
Gender	
Male	66 (47.14%)
Female	74 (52.86%)
Age category	
Year III	40 (28.57%)
Year IV	67 (47.86%)
Year V	33 (23.57%)
Ethnic Group	
Malay	96 (68.57%)
Chinese	13 (9.29%)
Indian	31 (22.14%)
Designation	
Civilian	67 (47.85%)
Cadet	55 (39.29%)
Territorial Army	18 (12.86%)

NB: data are presented with a number and the corresponding percentage in parentheses.

**Table 2 antibiotics-11-00540-t002:** Frequency and odds of gender in the formal education and training sufficiency and confidence in pharmacology.

	Male (n = 66)	Female (n = 74)	OR (95% CI)	*p*-Value
Commonly prescribed drugs				
Not sufficient	8 (36.4%)	14 (63.6%)	Ref.	
Sufficient	58 (49.2%)	60 (50.9%)	0.59 (0.23, 1.51)	0.273
Principles of appropriate and accurate prescription writing				
Not sufficient	17 (36.2%)	30 (63.8%)	Ref.	
Sufficient	49 (52.7%)	44 (47.8%)	0.51 (0.25, 0.99)	0.050
Infectious diseases				
Not sufficient	6 (35.3%)	11 (64.7%)	Ref.	
Sufficient	60 (48.8%)	63 (51.2%)	0.56 (0.20, 1.65)	0.301

NB: data are presented as a number with the corresponding percentage in parentheses and the odds ratio with a 95% confidence interval. Logistic regression was used to estimate the *p*-value.

**Table 3 antibiotics-11-00540-t003:** Frequency and odds of different student years in the formal education and training sufficiency and confidence in pharmacology.

	Y-III	Y-IV	Y-V	Y-III	Y-IV	Y-V	
Commonly prescribed drug							
Not sufficient	8 (36.4%)	13 (59.1%)	1 (4.55%)	Ref.	Ref.	Ref.	
Sufficient	32 (80.0%)	54 (80.6%)	32 (97%)	Ref.	1.04 (0.39, 1.77)	8.0 (1.05, 67.4)	0.049
Principles of appropriate and accurate prescription writing							
Not sufficient	17 (36.2%)	21 (31.3%)	9 (19.2%)	Ref.	Ref.	Ref.	
Sufficient	23 (25.6%)	46 (68.7%)	24 (26.7%)		1.62 (0.72, 3.63)	1.97 (0.73, 5.31)	
Infectious diseases							
Not sufficient	6 (15.0%)	10 (14.9%)	1 (3.00%)	Ref.	Ref.	Ref.	
Sufficient	34 (85.0%)	57 (85.1%)	32 (97.0%)		1.01 (0.34, 3.00)	5.64 (0.64, 49.4)	

NB: data are presented as a number with the corresponding percentage in parentheses and the odds ratio with a 95% confidence interval. Logistic regression was used to estimate the *p*-value. Y = Year.

**Table 4 antibiotics-11-00540-t004:** Frequency and odds of ethnicity in the formal education and training sufficiency and confidence in pharmacology.

	Malay	Chinese	Indian	Malay	Chinese	Indian
Commonly prescribed drug						
Not sufficient	19 (19.8%)	0	3 (9.68%)	Ref.	Ref.	Ref.
Sufficient	77 (80.2%)	13 (100%)	28 (90.3%)		-	2.32 (0.63, 8.41) (0.206)
Principles of appropriate and accurate prescription writing						
Not sufficient	34 (35.4%)	1 (7.69%)	12 (38.7%)	Ref.	Ref.	Ref.
Sufficient	62 (64.6%)	12 (92.3%)	19 (61.3%)		6.55 (0.82, 19.5) (0.076)	0.90 (0.63, 1.99) (0.740)
Infectious diseases						
Not sufficient	10 (10.4%)	3 (23.1%)	4 (12.9%)	Ref.	Ref.	Ref.
Sufficient	86 (89.6%)	10 (76.9%)	27 (87.1%)		0.39 (0.09, 1.65) (0.199)	0.79 (0.23, 2.72) (0.701)

NB: data are presented as a number with the corresponding percentage in parentheses and the odds ratio with a 95% confidence interval. Logistic regression was used to estimate the *p*-value.

**Table 5 antibiotics-11-00540-t005:** Frequency and odds of gender in the relative effectiveness of various modes of delivery of education instruction in gaining and retaining the knowledge of medicines and prescribing, including but not limited to antibiotics prescribing.

	Male (n = 66)	Female (n = 74)	Male	Female
Lectures				
Least useful	7 (10.6%)	8 (10.8%)	Ref.	Ref.
Useful	59 (89.4%)	66 (89.2%)		0.98 (0.34, 2.86) (0.969)
Tutorials, workshops or PBL sessions				
Least useful	6 (9.09%)	9 (12.2%)	Ref.	Ref.
Useful	60 (90.9%)	65 (87.8%)		0.72 (0.24, 2.16) (0.559)
Clinical rotations				
Least useful	1 (1.52)	2 (2.70%)	Ref.	Ref.
Useful	65 (98.5%)	72 (97.3%)		-
Informal teaching by fellow residents and registrars				
Least useful	8 (12.1%)	4 (5.41%)	Ref.	Ref.
Useful	58 (87.9%)	70 (94.6%)		2.41 (0.69, 8.50) (0.167)
Attending patient care rounds/ward rounds				
Least useful	2 (3.0%)	4 (5.41%)	Ref.	Ref.
Useful	64 (97.0%)	70 (95.6%)		0.55 (0.10, 3.10) (0.494)

NB: data are presented as a number with the corresponding percentage in parentheses and the odds ratio with a 95% confidence interval. Logistic regression was used to estimate the *p*-value. PBL = problem-based learning.

**Table 6 antibiotics-11-00540-t006:** Frequency and odds of student years in the relative effectiveness of various instructional modes in gaining and retaining the knowledge of medicines and prescribing, including but not limited to antibiotics prescribing (No significant differences were noted).

	Year-III	Year-IV	Year-V	Year-III	Year-IV	Year-V
Lectures						
Least useful	2 (5.0%)	11 (16.4%)	2 (6.06%)	Ref.	Ref.	Ref.
Useful	38 (95.0%)	56 (83.6%)	31 (93.9%)		0.27 (0.06, 0.79) (0.098)	0.82 (0.11, 6.11) (0.843)
Tutorials or workshops of PBL sessions						
Least useful	2 (5.0%)	13 (19.4%)	0	Ref.	Ref.	Ref.
Useful	38 (95.0%)	54 (80.6%)	33 (100%)		4.57 (0.97, 21.3) (0.054)	-
Clinical rotations						
Least useful	0	2 (2.99%)	1 (3.00%)	Ref.	Ref.	Ref.
Useful	40 (100.0%)	65 (97.0%)	32 (97.0%)		-	-
Informal teaching by fellow residents and registrars						
Least useful	2 (5.0%)	5 (7.46%)	5 (15.2%)	Ref.	Ref.	Ref.
Useful	38 (95.0%)	62 (92.5%)	28 (84.9%)		-	-
Attending patient care rounds/ward rounds						
Least useful	1 (2.50%)	4 (5.97%)	1 (3.00%)	Ref.	Ref.	Ref.
Useful	39 (97.5%)	63 (94.0%)	32 (97.0%)		-	-

NB: data are presented as a number with the corresponding percentage in parentheses and the odds ratio with a 95% confidence interval. Logistic regression was used to estimate the *p*-value.

**Table 8 antibiotics-11-00540-t008:** Frequency and odds of student years in confidence in prescribing knowledge.

	Y-III	Y-IV	Y-V	Y-III	Y-IV	Y-V
Pharmacology						
Somewhat confident	22 (55.0%)	42 (63.6%)	14 (42.4%)	Ref.	Ref.	Ref.
Confident	18 (45.0%)	24 (36.4%)	19 (57.6%)		0.57 (0.31, 1.55)	1.67 (0.66, 4.22)
Principles of prescribing						
Somewhat confident	30 (75.0%)	48 (71.6%)	14 (42.4%)	Ref.	Ref.	Ref.
Confident	10 (25.0%)	19 (28.4%)	19 (57.6%)		1.19 (0.49, 2.89)	4.06 (1.51, 11.02) (0.006)
Infectious diseases						
Somewhat confident	19 (47.5%)	27 (40.3%)	4 (12.1%)	Ref.	Ref.	Ref.
Confident	21 (52.5%)	40 (59.7%)	29 (87.9%)		1.34 (0.61, 2.94)	5.55 (1.95, 22.2) (0.002)

NB: data are presented as a number and the corresponding percentage in parentheses and the odds ratio with a 95% confidence interval. Logistic regression was used to estimate the *p*-value.

**Table 9 antibiotics-11-00540-t009:** Frequency and odds of race/ethnicity in the comparative confidence in knowledge.

	Malay	Chinese	Indian	Malay	Chinese	Indian
Pharmacology						
Somewhat confident	60 (63.2%)	6 (46.2%)	12 (38.7%)	Ref.	Ref.	Ref.
Confident	35 (36.8%)	7 (53.9%)	19 (61.3%)		1.99 (0.63, 6.42)	1.72 (1.17, 6.23) (0.019)
Principles of prescribing						
Somewhat confident	66 (68.8%)	7 (53.9%)	19 (61.3%)	Ref.	Ref.	Ref.
Confident	30 (31.3%)	6 (46.2%)	12 (38.7%)		1.88 (0.58, 6.11)	1.39 (0.60, 6.23)
Infectious diseases						
Somewhat confident	38 (39.6%)	7 (53.9%)	5 (16.1%)	Ref.	Ref.	Ref.
Confident	58 (60.4%)	6 (46.2%)	26 (83.9%)		0.56 (0.18, 1.80)	3.42 (1.21, 9.68) (0.021)

NB: data are presented as a number with the corresponding percentage in parentheses and the odds ratio with a 95% confidence interval. Logistic regression was used to estimate the *p*-value.

## Data Availability

Data is available only for research purpose from principal author (MH).

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
