# Peer review of "Antimicrobial Prescribing Confidence and Knowledge Regarding Drug Resistance: Perception of Medical Students in Malaysia and the Implications"

_antibiotics, 2022, doi:10.3390/antibiotics11050540_

Round 1

Reviewer 1 Report

In Lines 13 to 65, the authors' emails and ORCID ID, font type, and size should be revised to well-fit the body of the manuscript.

Line 67, the background section should be moved to the consequent line.

Line 67, more dangerous should be revised to medical terminology, for instance, resistant.

Line 69, physicians need to be trained inappropriate prescribing, starting with medical students, the sentence should be revised completely.

Line 71, drug resistance? The sentence needs to be completed, did the authors mean to discuss the drug resistance orientation?

Line 78, including infectious diseases? The sentence is incomplete and vague.

In lines 76 to 78, the authors should discuss whether according to a specific policy at their institute, medical students, and not yet fully registered and licensed physicians are capable of prescribing antibiotics.

The lack of coherence among lines 86 to 92, and 93 to 99 should be revised. The discussed global financial burden on the world bank scale seems far beyond the content of the article, and if at all needs to be discussed should be well narrated before lines 86 to 92.

Line 108, prudent should be replaced with another appropriate synonym to avoid duplicated use after line 107.

Lines 101 to 106, seems unnecessary explanation of the causalities for the topic choice. The content of the discussion mostly fits the epidemiological sections rather than the antibiotic section of the journal.

In lines 118 to 121, the authors mentioned an unethical act by HCP that is not easily defendable according to society's burdens. The points need to be explored more comprehensively.

Line 126: Inappropriate to be changed to unnecessary

Line 130: Would the author please clarify whether they meant self-treatment by medical professionals?

Line 130: The authors should clarify the antibiotics that are mentioned specifically. Otherwise, it would be both over the counter and prescription-mandatory antibiotics. The latter one would be an illegal act, and should be discussed in the context of professionalism.

Line 214: Years of student? The authors need to clarify whether they meant years at medical school

Line 226: Would the authors clarify what they specifically meant by response rate?

Line 245: What was the knowledge level criteria that was used?

Table 6: Could the authors clarify why to mention least useful? Whether they would rather replace with not helpful. In the current way, less useful, that would be misinterpreted with having a grade of usefulness and that quantification concept mandates further clarifications.

Author Response

Comments and Suggestions for Authors

In Lines 13 to 65, the authors' emails and ORCID ID, font type, and size should be revised to well-fit the body of the manuscript.

Author comments: Thank you – now revised.

Line 67, the background section should be moved to the consequent line.

Author comments: Thank you – now revised.

Line 67, more dangerous should be revised to medical terminology, for instance, resistant.

Author comments: Thank you – now revised.

Line 69, physicians need to be trained inappropriate prescribing, starting with medical students, the sentence should be revised completely.

Author comments: Thank you – now revised.

Line 71, drug resistance? The sentence needs to be completed, did the authors mean to discuss the drug resistance orientation?

Author comments: Thank you – now revised.

Line 78, including infectious diseases? The sentence is incomplete and vague.

Author comments: Thank you – now revised.

In lines 76 to 78, the authors should discuss whether according to a specific policy at their institute, medical students, and not yet fully registered and licensed physicians are capable of prescribing antibiotics.

Author comments: Thank you – now revised.

The lack of coherence among lines 86 to 92, and 93 to 99 should be revised. The discussed global financial burden on the world bank scale seems far beyond the content of the article, and if at all needs to be discussed should be well narrated before lines 86 to 92.

Author comments: Thank you. We have now updated this section to discuss first the high and growing number of annual deaths due to AMR before discussing the economic impact. Both are important as seen by the appreciable efforts worldwide to prevent/ improve the management of patients with COVID-19. The clinical and economic consequences have been a key stimulus behind WHO and National Action Plans to reduce AMR – as now documented. This includes Malaysia in view of growing AMR rates and inappropriate prescribing. We hope this is now acceptable as we believe the financial implications of AMR are a critical issue going forward. 

Line 108, prudent should be replaced with another appropriate synonym to avoid duplicated use after line 107.

Author comments: Thank you – now revised.

Lines 101 to 106, seems unnecessary explanation of the causalities for the topic choice. The content of the discussion mostly fits the epidemiological sections rather than the antibiotic section of the journal.

Author comments: Thank you for this. We have revised these sections to improve the flow and hope this is now acceptable.

In lines 118 to 121, the authors mentioned an unethical act by HCP that is not easily defendable according to society's burdens. The points need to be explored more comprehensively.

Author comments: Thank you – we have now revised this section and hope this is clearer and OK.

Line 126: Inappropriate to be changed to unnecessary

Author comments: Thank you – now revised.

Line 130: Would the author please clarify whether they meant self-treatment by medical professionals?

Author comments: Thank you – now revised.

Line 130: The authors should clarify the antibiotics that are mentioned specifically. Otherwise, it would be both over the counter and prescription-mandatory antibiotics. The latter one would be an illegal act, and should be discussed in the context of professionalism.

Author comments: Thank you – now revised with additional references. We hope this is now OK

Line 214: Years of student? The authors need to clarify whether they meant years at medical school

Author comments: Thank you – now revised.

Line 226: Would the authors clarify what they specifically meant by response rate?

Author comments: Thank you – now revised.

Line 245: What was the knowledge level criteria that was used?

According to the QUESTIONAIRE [(8.2) Principals of Appropriate and accurate prescription writing] Year-V students responded that they possess “more than sufficient knowledge”.  Additionally, our estimated odds ratio(OR) was 8. Thus we mentioned Year-V students had higher knowledge than others.

Table 6: Could the authors clarify why to mention least useful? Whether they would rather replace with not helpful. In the current way, less useful, that would be misinterpreted with having a grade of usefulness and that quantification concept mandates further clarifications.

Many Thanks. This research maintained ORIGINAL terminology of Questionnaire developed by  Weier N, Thursky K, Zaidi STR. Antimicrobial knowledge and confidence amongst final year medical students in Australia. PLoS One. 2017;12(8):e0182460. Published 2017 Aug 3. doi:10.1371/journal.pone.0182460

Reviewer 2 Report

   This manuscript describes the importance of education for the medical students to learn antibiotic resistance. Similar reports have been done from several countries in the world and this is from Malaysia, which seems to be beneficial. However there remains some points to be reconfirmed.

Major points

   In this study, the authors compared knowledge between genders and between races. But is it important? I don’t think gender or race (especially gender) is important for learning the appropriate use of antibiotics.

Minor points

  1. In Table 1, the authors show Age category, 21 years, 22 years and 23-24 years. But in other tables, they describe Year-III, Year-IV and Year-V, which seems to be better. Please unify the description. Grades look more important than years.
  2. In Figure 5, the X scale display of the graph seems to be shifted to the left.
  3. Please describe what the upward and downward triangles, circles, squares, etc. indicate.
  4. As to Figure 5-7, please make it easy to understand, such as by adding a legend. 

Author Response

This manuscript describes the importance of education for the medical students to learn antibiotic resistance. Similar reports have been done from several countries in the world and this is from Malaysia, which seems to be beneficial. However there remains some points to be reconfirmed.

Author comments: Thank you for these comments. We hope we have satisfactorily addressed these.

Major points

In this study, the authors compared knowledge between genders and between races. But is it important? I don’t think gender or race (especially gender) is important for learning the appropriate use of antibiotics.

Author comments: Thank you for this comment. However – may we beg to differ. We have found these issues important in previous studies we have undertaken together with others which we have referenced. Consequently, we have kept this in. We hope this is OK with you. Additionally, both sex and race had influence the estimate thus those was used as covariates in others regression model.

Minor points

  1. In Table 1, the authors show Age category, 21 years, 22 years and 23-24 years. But in other tables, they describe Year-III, Year-IV and Year-V, which seems to be better. Please unify the description. Grades look more important than years.

Thanks Sir for valuable comment. We have altered.

  1. In Figure 5, the X scale display of the graph seems to be shifted to the left.

Thanks. We have altered Figure 5.

  1. Please describe what the upward and downward triangles, circles, squares, etc. indicate.

Figure 1: Circles indicates the estimate of odds ratio.

Figure 2 & 6: It was given in the figure that Circle indicates III year students, square indicated IV years Students and triangles indicate V years’ students.

Figure 3 & 4: Triangle and circle the estimates of Odds ratio of IV and V year student respectively.

Figure 8: Circles are indicating the estimate of odds ratio.

  1. As to Figure 5-7, please make it easy to understand, such as by adding a legend. 

Thanks Figure legends was modified in the main text.

Round 2

Reviewer 1 Report

  • Considering the race, the following are considered standard racial subcategories. However, in table 1, the authors have categorized the study population in a different manner. 
  • White or Caucasian - British, French, German, etc.
  • Black - Kenyan, Nigerian, Somalian, biracial, etc.
  • American Indian or Alaska Native - Iroquois, Cherokee, Navajo, Haida, etc.
  • Latino or Hispanic - Cuban, Mexican, Puerto Rican, etc.
  • Asian - Japanese, Korean, Chinese, Cambodian, etc.
  • Pacific Islander or Hawaiian - Samoan, Tongan, Maori, Tahitian, etc.

Author Response

Comments and Suggestions for Authors

  • Considering the race, the following are considered standard racial subcategories. However, in table 1, the authors have categorized the study population in a different manner. 
  • White or Caucasian - British, French, German, etc.
  • Black - Kenyan, Nigerian, Somalian, biracial, etc.
  • American Indian or Alaska Native - Iroquois, Cherokee, Navajo, Haida, etc.
  • Latino or Hispanic - Cuban, Mexican, Puerto Rican, etc.
  • Asian - Japanese, Korean, Chinese, Cambodian, etc.
  • Pacific Islander or Hawaiian - Samoan, Tongan, Maori, Tahitian, etc.

Thanks Sir for valuable comment. We are sorry to mention RACE it should be Ethnic Group. Malaysia have THREE major ethic group – Malay, Chinese, and Indian. We have altered.

Reviewer 2 Report

I confirmed that the manuscript has been revised. 

Author Response

Thanks, Sir for your kind comment on our paper.
